

# 1 Data flow of spectral UV measurements at Sodankylä and 2 Jokioinen

**J.S. Mäkelä[1*], K. Lakkala[1,2], T. Koskela[1], T. Karppinen[2], J.M. Karhu[2] , V. Savastiouk[4],** 5 **H. Suokanerva[2], J. Kaurola[1], A. Arola[3], A.V. Lindfors[1], O. Meinander[1], G. Leeuw[1,5] and** 6 **A. Heikkilä[1]**
[1] {Finnish Meteorological Institute, Research and Development, 00101 Helsinki, Finland}
[2] {Finnish Meteorological Institute, Sodankylä, Finland}
[3] {Finnish Meteorological Institute, Kuopio, Finland}
[4] {IOS Inc, Toronto, Canada}
[5] {Department of Physics, University of Helsinki, 00014 Helsinki Finland}
[*]{now at: University of Jyväskylä, Jyväskylä, Finland}
Correspondence to: K. Lakkala (Kaisa.Lakkala@fmi.fi)
**Abstract**
We describe the steps that are used at the Finnish Meteorological Institute (FMI) to process 17 spectral ultraviolet (UV) radiation measurements made with its three Brewer 18 spectrophotometers, located in Sodankylä (67°N) and Jokioinen (61°N). The spectrum is 19 measured many times a day, following a pre-programmed schedule. Multiple corrections are 20 made to the data in near real time (dark current, dead time, stray light, noise spikes, 21 temperature, and cosine correction) and quality control is also performed automatically. The 22 Brewers are integrated into the operational control systems at FMI, allowing both quick 23 responses to malfunctions and quick dissemination of the data products. Several data products 24 are produced, including the near-real-time UV index and various daily dosages. The daily 25 doses are calculated each morning for the previous day's data. Once per year the responses of 26 the Brewers are recalculated, and the corrected data are uploaded to the European UV 27 database hosted by FMI.



## 29 1 Introduction


The Brewer spectrophotometer (Brewer) (Bais et al. 1996; Brewer, 1973) is originally
degined to measure total ozone, but has also been developed to measure the spectral UV
irradiance and sulfur dioxide ($SO_2$). At present there are over 220 instruments set up by
research institutes all over the world (http://kippzonen-brewer.com/). These instruments form
an important network for monitoring changes in the total ozone column and, e.g., are used as
validation measurements for satellite retrievals. The Finnish instruments were set up in 1990
and 1995, in Sodankylä and Jokioinen, respectively, to respond to the need to monitor total
ozone and UV radiation after the discorvery of the Arctic ozone loss. Nowadays, these
spectral UV time series of over twenty years are unique and among the longest measured in
the Arctic. The homogenized time series have been used for several international studies
related to Arctic ozone loss (e.g., Bernhard et al. 2013; Manney et al. 2011; Knudsen et al.
1998), satellite data validation (Hassinen et al. 2008), biological (e.g., Lappalainen et al.
2010; Martz et al. 2009), material (Heikkilä 2014) and health research (Kazantzidis et al.

44 2009).


The high dynamical range of UV radiation reaching the surface of the Earth sets challenges
to the instruments, which are designed to monitor both the short UV-B wavelengths (290-315
nm) and the longer UV-A wavelengths (315-400 nm). Also the Brewer is a versatile and an
extremely complex instrument, with many intermediate steps and corrections in the
processing chain from data acquisition to final data dissemination. High quality data can only
be ensured after careful characterization of the instrument, correction of known measurement
errors and careful quality control (QC) and quality assurance (QA). (Seckmeyer et al. 2001;
Garane et al. 2006; Lakkala et al. 2008; Webb et al. 2003).

In particular, keeping a Brewer absolutely calibrated is difficult (Bernhard and Seckmeyer
1999, Webb et al. 1998). International campaigns are organized to evaluate the calibration and
measurement procedures performed by different Brewers and institutes. The difficulty of the
absolute calibration was seen in the last European Brewer comparison organized by the COST
1207 project in El Arenosil, Spain. There, 6 Brewers from 18 differed more than 10% from
the reference, when using the calibration provided by the operator. During the comparison
campaign, each instrument was recalibrated against a common calibration lamp. The



62calibration procedure was performed by only one operator. The results showed that the

63difference between the two calibrations could be even more than 20%. When using the

64calibration based on the common lamp, the difference between most of the Brewers

65diminished to be within ± 6% (Julian Gröbner, personal communication). The remaining

66difference could result from different data correction and data processing procedures, e.g.,

67differences in the way to take into account the temperature dependence and the angular

68response of the instrument.

69

70To enable Brewer data from around the world to be comparable, it is necessary to very

71carefully document the traceability of the calibration and how the data has been processed.

72Careful documentation should be part of routine QC/QA procedures at each site. This allows

73anyone to audit all steps which have been taken before delivering the data, and allows

74changes to be made in post-processing without starting everything from the beginning.

75

76This paper documents the steps that are involved in the acquisition, processing, storage, and

77dissemination of data from the Finnish Brewers, located in Sodankylä and Jokioinen. The

78observatory at Jokioinen is in the process of being shut down, and the spectral UV

79measurements have been moved to Helsinki. Thus, this paper also serves as a historical

80description of the Jokioinen measurements. A detailed description will be given of the process

81flow from the raw photon counts to the calibrated spectral UV irradiances and UV products.

82We also describe the quality control and quality assurance systems that are used to ensure

83valid output. In a companion paper (Mäkelä et al. 2015, this issue) we describe how the

84calibration and homogenization of the data is made. In another companion paper (Heikkilä et

85al. 2015, this issue) we describe how the data are further processed in the EUVDB database.

86

872  **Description of the stations**

88In Brewers of the Finnish Meteorological Institute (FMI) are operated at two sites: Sodankylä

89and Jokioinen. Below we briefly describe the site characteristics and which instruments are

90used at each site.





## 91 2.1    2.1 The Sodankylä station (Brewer #037 and #214)

The Finnish Meteorological Institute's Arctic Research Centre is located at 67.37°N, 27.63°E,
altitude 179 m above sea level, in Sodankylä. It has  had an operating Brewer  Mark II since
1990. The station is described e.g. in Lakkala et al 2003. The near surroundings are pine
forest, and at the South-West flows the river Kitinen. The area is surrounded by a large
peatland area in the East. There is snow cover from October to late April. The sun is just
below the horizon from mid-December to mid-January.  Temperatures are ranging from -40C
in winter to +30C in summer.
Two Brewers are currently operated at this site: #037 and #214. They are located on the roof
of the sounding station (see Figure 1). Brewer #037 has a single monochromator and the
wavelength range is 290-325 nm. The instrument has a Teflon diffuser and the width of the
slit function is 0.56 nm at FWHM.  the later Brewer, #214, has been set up in 2012,  in order
to work in tandem with Brewer #037 and measure the longer wavelength part of the UV-
radiation, as it measures the UV spectrum up to 365 nm. The slit function of the new Brewer
is 0.55 nm at FWHM.

## 106 2.2    2.2 The Jokioinen station (Brewer #107)

The meteorological observatory in Jokioinen (60.82°N, 23.50°E) is at an altitude of 107 m
above sea level. The observatory is located  in a rural area surrounded by fields and mainly
coniferous forest. The ground is covered by snow most of the time during December-March.
Temperatures can range from -20C to +30C. The observatory will be shut down in the near
future and the Brewer was moved to Helsinki in November 2015. The Brewer was located on
the roof of the sounding station (see Figure 2 and Figure 3).
The FMI acquired the current Mk III Brewer #107 in the observatory in Jokioinen in 1995.
Brewer #107 has a double monochromator.  It collects UV radiation with a hemispherical
field-of-view through a PTFE diffuser enclosed in a quartz dome. It originally operated in the
wavelength range 286.5-363 nm, but the optics were changed in April 1997, and its
wavelength range since then is 286.5-365- nm. The width of the slit function is 0.59 nm at
FWHM.



**3 Data processing**
**3.1 Obtaining raw data**
Figure 4 shows the main data flow of the Brewers at FMI. The Brewers have their own
operating computers, but the processing of all the data from both stations is done at a single
central Unix server. The central computer checks the operational computers every 5 minutes.
Any new data are uploaded, and UV data processing begins. The data products are stored at
the central computer. Applications that use the data products (see Section 3.4) may access the
central computer at any time and are given the latest valid data.

Each Brewer is associated with a nearby automatic weather station (AWS) which measures
the visibility according to its own schedule (currently every ten minutes). This information
and the total ozone calculated by the Brewer is used in the cosine correction procedure. As
auxiliary measurements, a broadband UVB radiometer (Solar Light SL501A) and CM11
pyranometer (Kipp & Zonen), which measures global radiation (305–2800 nm), are
synchronized to the Brewer measurements. The measurements of the operational SL501A
radiometer of the site are used in non-realtime quality assurance procedures to identify
erroneous measurements.

The Brewer operational computers work autonomously, and make measurements based on
schedules that are predefined by the operator. Over the years, a variety of schedules have been
used, and the schedules for the two sites have supported slightly different research targets at
different times. For example, at Jokioinen, more frequent measurements have been made near
sunrise and sunset, and at constant air masses, as well as at time of the smallest solar zenith
angle (SZA). At Sodankylä, measurements have generally been spread out more evenly
throughout the day. However, in terms of the long-term statistical quality of the data, these
differences  do not matter. At both sites, there has been a measurement at least every half hour
during daylight (which can be short in the winter, in particular in Sodankylä due to its
location just north of the Artic circle) and a measurement at midday.  The exact number of
measurements at Jokioinen, for example, may range from about 8 in winter to more than 30 in
summer.




The stability of the Brewer is monitored by measuring an internal lamp (typically about 2-5
times per day, see Figure 5) at the six wavelengths used for total ozone retrieval. The
information about the stability is used for the post processing of total ozone measurements,
during which the effect of changes in the instrument can be corrected. In addition, every three
weeks a more extensive stability check is made using external 50 W lamps. Then, the whole
UV wavelength range is measured, and possible drifts in the spectral response of the Brewer
can be detected and corrected afterwards during the post processing of UV spectra.
**3.2    Processing of spectral data**

The spectral processing is done using custom-made software, which is based on the original
software provided by the manufacturer, mostly perl and shell scripts. In addition to the near-
real-time measurements, every morning the data from the previous day is reprocessed and
checked, and all relevant databases updated.  The algorithms have been described in detail by
Lakkala et al. (2008), and only key points will be summarized here (see Figure 6).

Scans are performed from small to larger wavelengths. The Brewer also returns the dark
current for each scan. The total scanning time is about 4 minutes for Brewer #037 and 5
minutes for Brewers #107 and #214 due to the larger wavelength range. The noise spikes are
removed based on the method of Meinander et al. (2003). The dark current count is subtracted
from measured counts. The dead time is measured daily, and the data are corrected using an
iteration of an exponential function including the number of counts and the dead time values.
The stray light is calculated as the average of all counts below 292 nm (#107 and #214) or
293 nm (#037). Since #107 and #214  have a double monochromator, the stray light counts
are small, while the Sodankylä Brewer #037 has a single monochromator, and the stray light
counts are larger (Bais et al. 1996).

The counts are then converted to irradiances by dividing the counts by the daily response. The
determination of the daily response is briefly described in Lakkala et al. (2008) and is covered
in more detail in the companion paper  (Mäkelä et al. 2015, this issue). The temperature and
cosine corrections are then made to the spectral irradiances.



## 181 3.3 Data products

The Brewer UV spectra are used to calculate dose rates and daily doses using at least the 184 following action spectra:

- Erythema (CIE weighting function, McKinlay et al. 1987)
- Skin cancer in mice corrected for human skin, 299 nm normalization (de Gruijl, F.R. 187 and J.C. van der Leun, 1994)
- UVB non weighted 290-320 nm
- UVB non weighted 290-315 nm
- UVA non weighted 315-400 nm
- Generalized plant response (normalised at 300 nm) (Caldwell et al. 1986)
- DAN damage (Setlow 1974)
- Photosynthesis inhibition (Mitchell 1990)
- Spore dosimetry AS SIDv2.2 (Munakata et al. 2000)
- Previtamin D (CIE 2006)

In addition the UV index is calculated by multiplying the CIE erythemally weighted UV 198 irradiance in $Wm^{-2}$ by 40 (WMO 1997).

Data are uploaded to the following databases.

- The IDEAS database for quick and long-term quality control: every five minutes
- The FMI climate database (UVI index) every time a new UVI measurement is made.
- The EUBREWNET database for collaboration with the international Brewer 203 community within the COST 1207 project: every 20 minutes 204 (http://rbcce.aemet.es/eubrewnet)
- The database of the FMI-Arctic Research Centre (http://litdb.fmi.fi/): Sodankylä data, 206 once a year
- The European UV database (http://uv.fmi.fi/uvdb/): once a year

## 209 3.4 IDEAS: Real-time QA and monitoring

In 2015 a new software has been introduced to facilitate both quick and long term quality 211 control of data, and to improve the potential of the Brewers to work as real-time operational 212 devices. IDEAS is a tool for cheking that the Brewer is functioning correctly. The Brewer 213 itself makes several check measurements during the day, and these measured parameters are 214 used to monitor the stability of the instrument.



The IDEAS software is also used e.g. to calculate the daily mean of total ozone, which can be
directly submitted to databases. Every measurement of the Brewer and process of the
operating software is recorded in addition to appropriate data files. These are stored as so-
called B-files, and  updated to the  server in which IDEAS is running.

Two sample screenshots are shown (Figure7   and Figure 8) as an example of possible
warnings and total ozone calculation and comparison with satellite (OMI-instrument) total
ozone.   IDEAS is integrated with the real-time 7/24 operational control system of FMI.
Automatic warnings of malfunctions of the Brewer are generated within 20 minutes and sent
to the control center. If the personnel there are unable to solve the problems,  stand-by Brewer
specialists can be alerted by text messages when needed.

**4    Annual quality assurance**

The responsivity times series of the Brewers are recalculated typically once a year. During the
process, the drifts of the instruments or the calibration lamps are taken into account. The
spectra are recalculated with the methods described in a companion paper (Mäkelä et al 2015,
this issue). If necessary, a wavelength correction is made using the SHICRIVM algorithm
(Slaper et al 1995). The dose rates are compared with reconstructed UV, model calculations
of clear sky UV and global radiation, global radiation and broad band UV data in order to
distinguish erroneous measurements. In addition each spectrum is checked by eye and bad
measurements are excluded.

**5    Conclusions**
The FMI has operated Brewer spectrophotometers since the 1990's in two locations,
Jokioinen and Sodankylä. During that time, FMI has implemented all corrections and
improvements that have been identified by the Brewer community. The outputs are used to
calculate multiple UV products which require spectral data. A special new focus is being put
on real-time quality control, with the IDEAS software allowing warnings of malfunctions to
be sent to operators very rapidly. Although the measurements at the Jokioinen station has



245been stopped, Sodankylä will continue to provide one of the longest continuous spectral UV
246time series in the world, and new time series will start in Helsinki.


**Acknowledgements**

249Professor Esko Kyrö is acknowledged for starting the Brewer measurements at Sodankylä.
250We are grateful to the operators at Sodankylä and Jokioinen stations for daily maintenance
251and for performing the calibrations of the Brewers. We thank the Brewer community within
252the COST 1207 project for sharing expertise related to Brewer measurements.

253

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






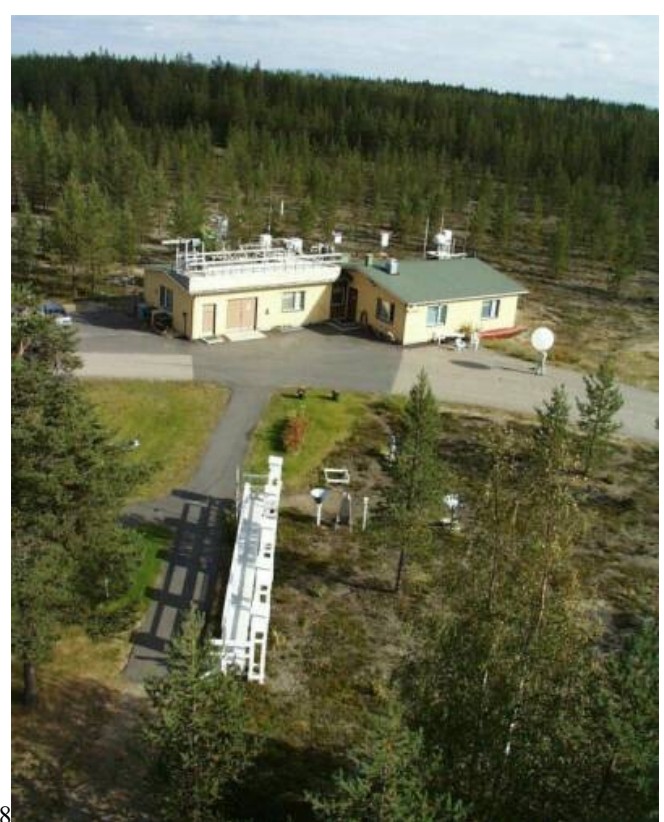

359Figure 1. View of Sodankylä observatory. The Brewer is located on the roof in the enclosure on the left.




361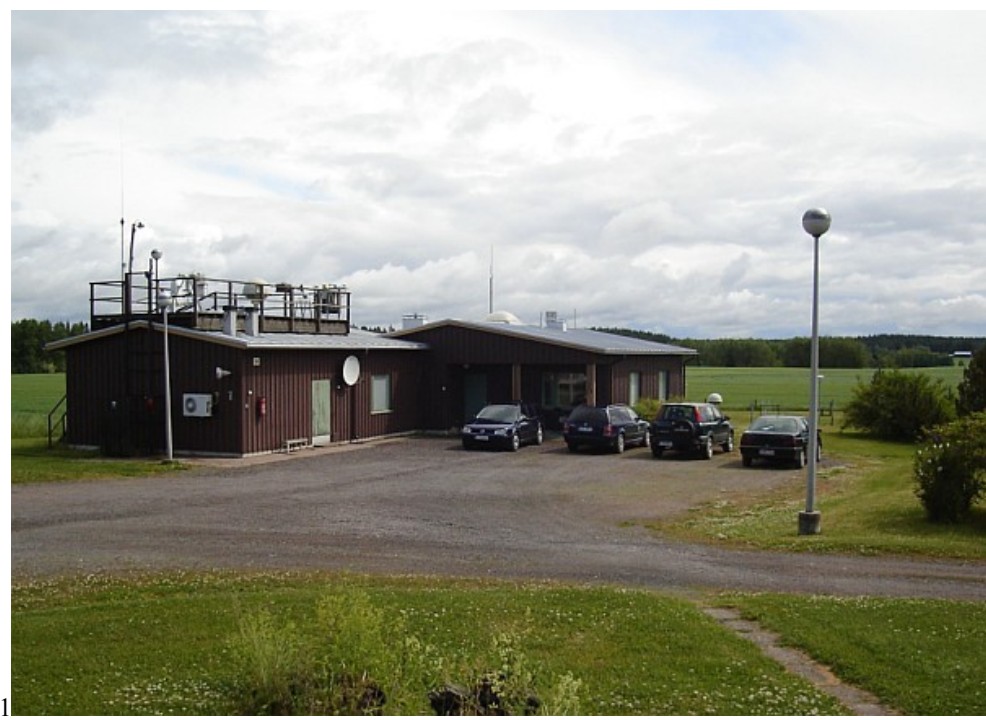


363Figure 2. View of Jokioinen observatory. The Brewer is located on the roof in the enclosure on the left.


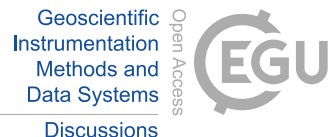

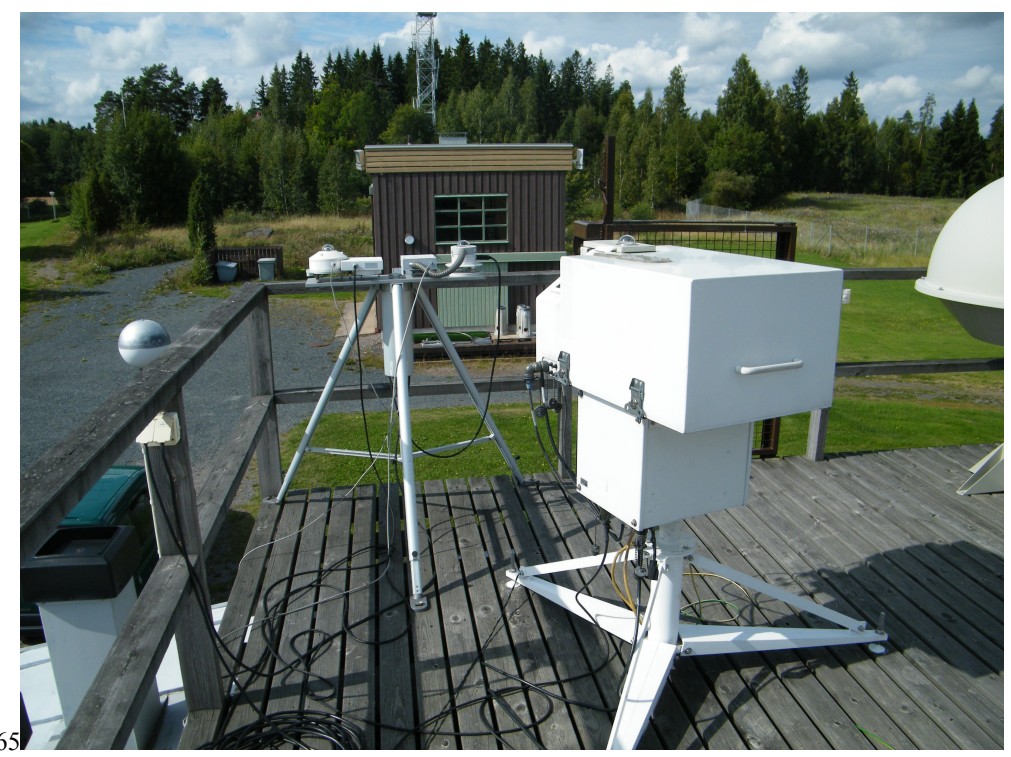


366Figure 3. Close up of Brewer # 107 on the roof of the Jokioinen observatory.






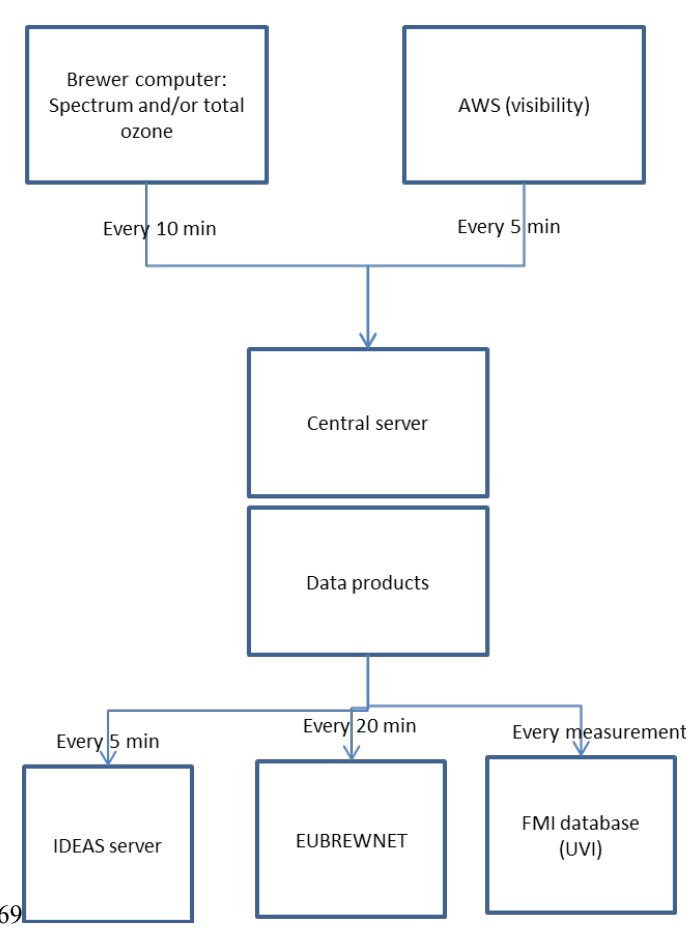



Figure 4. Main data flow of Brewer measurements at the FMI.











**Standard lamp R6 (0 sec)**

| File name | Count | Serial | Expected (+/-20) | Measured | 30-day history |
|---|---|---|---|---|---|
| B25915.037 | 12 | #037 | 1670 | 1874 | |
| B25915.107 | 10 | #107 | 470 | 452 | |
| B25915.214 | 13 | #214 | 240 | 236 | |



Figure 5. The stability of the Brewers can be traced e.g. by using the R5 and R6 values of the internal standard
lamp (output from IDEAS).






A few times a day

**Measure dead time**

On every scan

**Measure dark current**

**Scan in upward direction**

**Remove noise spikes**

**Correct dark current**

After darkroom calibrations (every 6 weeks). See companion

**Recalculate responses**

**Correct for dead time**

**Correct for stray light**

**Convert to irradiance**

**Make temperature correction**

When updated

**Get auxiliary data**

**Make cosine correction**

**Transfer to server**


Figure 6. Detailed data flow of Brewer UV measurements at FMI.





**Critical warnings from b-files' comments for days between 2015257 and 2015259 (1 sec, 10 rows)**

| Type | Comment | count | priority | fname | location | serial | time | value | min | max |
|------|---------|-------|----------|-------|----------|--------|------|-------|-----|-----|
| fz: | Suppressed oscillation of filter wheel. | 7 | 5 | B25715.107 | Jokio | #107 | 05:04:32 | 1 | -1 | |
| fz: | Suppressed oscillation of filter wheel. | 7 | 5 | B25815.107 | Jokio | #107 | 05:33:54 | 1 | -1 | |
| fz: | COUNTS TOO HIGH - aborting | 6 | 5 | B25815.107 | Jokio | #107 | 05:35:05 | 1 | -1 | |
| ds: | DS intensity too low ( 13/cy), skipping. | 9 | 99 | B25815.037 | Sodan | #037 | 05:39:58 | 1 | -1 | |
| ds: | DS intensity too low ( 23/cy), skipping. | 30 | 99 | B25715.037 | Sodan | #037 | 05:41:44 | 1 | -1 | |
| ds: | DS intensity too low ( 28/cy), skipping. | 16 | 99 | B25915.037 | Sodan | #037 | 05:43:34 | 1 | -1 | |
| ds: | DS intensity too low ( 54/cy), skipping. | 9 | 99 | B25715.214 | Sodan | #214 | 05:38:43 | 1 | -1 | |
| ds: | DS intensity too low ( 77/cy), skipping. | 11 | 99 | B25915.214 | Sodan | #214 | 05:46:57 | 1 | -1 | |
| rl: | 2015 257 22 21 02 New value set for watermark LOOP.RUN.TIME: 699.000025 millisec. | 6 | 5 | B25815.214 | Sodan | #214 | 22:09:48 | 1 | -1 | |
| rl: | 2015 256 22 28 01 New value set for watermark LOOP.RUN.TIME: 673.000050 millisec. | 7 | 5 | B25715.214 | Sodan | #214 | 22:10:07 | 1 | -1 | |


Figure 7. Sample output from IDEAS software. Critical warnings. This information can be sent to operators in
real time.

**Ozone comparison with OMI for the last week (0 sec, 21 rows)**

| Location | Serial | Date | YYYYJJJ | OMI ozone | Brewer ozone | Percent difference | Cloudiness range (of 8) | Cloudiness most of the day | Average T | Visibility range, km | Average pressure, hPa |
|----------|--------|------|---------|-----------|--------------|--------------------|-----------------------|----------------------------|-----------|----------------------|------------------------|
| Jokioinen | 107 | 2015-09-15 | 2015258 | 291+/-3 | 294+/-1.8 | -1.0 | 0-0 | 0 | 16.1 | 20-20 | 1014 |
| Sodankyla | 037 | 2015-09-15 | 2015258 | 290+/-4 | 299+/-1.3 | -3.0 | 0-0 | 0 | 15.0 | 20-20 | 1020 |
| Sodankyla | 214 | 2015-09-15 | 2015258 | 290+/-4 | 306+/-1.3 | -5.2 | 0-0 | 0 | 15.0 | 20-20 | 1020 |
| Jokioinen | 107 | 2015-09-14 | 2015257 | 292+/-4 | 298+/-2.5 | -2.0 | 0-0 | 0 | 16.0 | 20-20 | 1024 |
| Sodankyla | 037 | 2015-09-14 | 2015257 | 292+/-4 | 301+/-3.4 | -2.9 | 0-0 | 0 | 14.6 | 20-20 | 1024 |
| Sodankyla | 214 | 2015-09-14 | 2015257 | 292+/-4 | 310+/-3.1 | -5.8 | 0-0 | 0 | 14.6 | 20-20 | 1024 |
| Jokioinen | 107 | 2015-09-13 | 2015256 | 299+/-3 | 300+/-2.7 | -0.3 | 0-0 | 0 | 16.1 | 20-20 | 1029 |
| Sodankyla | 037 | 2015-09-13 | 2015256 | 290+/-2 | 294+/-3.6 | -1.3 | 0-0 | 0 | 15.1 | 20-20 | 1030 |
| Sodankyla | 214 | 2015-09-13 | 2015256 | 290+/-2 | 303+/-3.2 | -4.2 | 0-0 | 0 | 15.1 | 20-20 | 1030 |
| Jokioinen | 107 | 2015-09-12 | 2015255 | 293+/-5 | 302+/-5.9 | -2.9 | 0-0 | 0 | 16.9 | 11-20 | 1035 |


Figure 8. Sample output from IDEAS software. Comparison of measured ozone to reference ozone from OMI
satellite.