# Peer review of "Data flow of spectral UV measurements at Sodankylä and 2 Jokioinen"

_Geoscientific Instrumentation, Methods and Data Systems, 2015_

## Referee Comment (RC1) · Anonymous Referee #1 · 1 Feb 2016

Review of the manuscript entitled: "Data flow of spectral UV measurements at Sodankylä and Jokioinen"

From my point of view the manuscript deals with the methodology that is followed by FMI scientists in order to measure, quality assure, post process and submit solar UV data related with two instruments that have been provided two of the longest UV data series worldwide. So the idea itself of reporting this methodology is worth publishing.

However, the paper itself does not particularly describe or properly reference all the methodology involved from the raw measurements till the final submission of the post processing data. It has structural problems and the link on the methods used – actual data and their presentation is weak.

Major comments
The English language used throughout the paper needs major improvements. There are various cases that words are missing or sentences are difficult to understand.

Introduction

Paragraph 1

"The Brewer was designed to measure ozone but then UV and now SO2 is measured". The correct way to write that, is that the instrument was initially designed to measured total column ozone with the differential absorption method (reference) using the direct sun port. In addition, the global (diffuser) port was introduced for measurements of spectral UV. Then the use of the direct sun data have used in order to calculate SO2 (reference), Aerosol Optical depth (e.g. Groebner, Kazadzis, Marenco ..), NO2 (e.g. Cede, Diemoz,..)

Paragraph 2

It is interesting to describe this dynamic range and in addition some details about the difficulty for such measurements for places like e.g. Sodankyla where low solar elevations is challenging for UV measurements.

Paragraph 3

It is interesting to mention a recent intercomparison campaign. But these results are not yet published. So in addition to this you could add results of a large number of previous publications that report such differences. (for example the SUSPEN paper by A. Bais and various others also in form of EU publications related with the QASUME travelling standard). In addition, describing this recent campaign someone has to clarify if this 20% down to 6% calibration differences are due to: primary calibration sources differences and processing differences or (/and) changes in the instruments due to their travel to the campaign site.

Last paragraph of the introduction
[Figure]

You are mentioning steps like: acquisition, processing, storage, and dissemination. I think that a proper way to describe all these steps is to start from :

a. Acquisition and Quality control of raw or e.g. level 0 data.

b. Calibration of the measured data

c. Quality control and corrections (spikes, straylight, dead time, etc) towards e.g. level 1 data.

d. Online real time checks

e. Post processing of spectral data and creation of e.g. level 2 or metadata.

I will get back to this point in the following sections

Section 2

It would be useful to provide some graphical example of the data availability from the beginning of the measurements till today. Also a simple graph showing the calibration record (e.g. instrument responses) for all the period, accompanied with the calibration uncertainties that can be probably differ from the early 90s till today.

I think that 3 pictures of the brewer locations can be only two, one from Sodankyla and one from Jokioinen.

Section 3

In order to make all the individual steps clear, I would stick to a format similar than the one described in the end of my comments for paragraph 1. It is important to make the individual steps very clear as this is all that the paper is about.

A table with the initial steps is essential. But more essential is to accompany this table with the references describe each of the steps. (e.g. figure 6 could be modified towards this goal) Figure 5 . Instead of providing a print screen image here. You can show internal lamp time series and examples of good and bad measurements and
corrections

3.3. Some acronyms are not defined

It is not clear if the IDEAS software is applied in the raw data or in the calibrate & corrected ones. I think you have to report that the software assures of specific "bad data" or instrument failures and not an evaluation of the post correction or calibration methods that are applied.

Instead of screenshots of "possible warnings" for describing the QA/QC you have to report on all warnings that are included in the process.

Section 4

I do not understand why "if necessary a wavelength correction is made." How do you know that is necessary or not if you do not already have a report on possible wavelength shifts ?

Personal remark

Copying from the manuscript:

"These instruments form an important network for monitoring changes in the total ozone column . . .. Nowadays, these spectral UV time series of over twenty years are unique and among the longest measured in the Arctic. "

and

"The observatory at Jokioinen is in the process of being shut down, and the spectral UV measurements have been moved to Helsinki. Thus, this paper also serves as a historical description of the Jokioinen measurements."

So Jokioinen is a "unique" and for sure among the longest not only in the arctic but worldwide, station for ozone and UV measurements but it is in the process of being shut down. Since this paper is co-authored and co-signed from 12 FMI scientists, I

want to pose the question: how it is possible to refer it here as a unique and historical station and at the same time to shut it down? And of course moving the instruments somewhere else is not an ideal way of a continuation of this 20 year series.

So for me it seems at least ironic to say that this paper is important (also) because Jokioinen is a unique station in terms of time series. Now, a paper that would show a full 20 year time series of Jokioinen Brewer measurements, calibration efforts, e.t.c. sure would serve as a historical description. On the other hand describing procedures for online quality control (only recently added) for a station that closes down are not useful for a future user of the data, as much as a report/publication on the actual calibrations, correction procedure, uncertainty analysis and measurement trend analysis for the particular station.
* * *

---

## Referee Comment (RC2) · Anonymous Referee #2 · 7 Feb 2016

This paper does not provide any scientifically sound results. It contains a pure outline of the procedures followed for the UV measurements at the two FMI stations but it does not contain any details of the methods or algorithms used in these procedures, or any data. In its present form, it can serve only as a source of general information, like a web site. It is true that detailed descriptions of sites and procedures and different steps in the data processing chain have been published elsewhere, but just a listing of procedures and referencing other papers cannot justify the publication of the manuscript. I think it must be further expanded and structured in a way that could be useful and applicable to other, for example newly established, stations. Furthermore, I think that it could be useful to show at least some sort of time series with measurements at the two sites.

Specific comments

59-68: Are these statements supported only by "Groebner personal communication"? There have not been published elsewhere?

131: Is there any reference where the cosine correction procedure is described?

131: AWS sampling is ten minutes but the flowchart in Figure 4 sates 5 min. Which of the two is correct?

136: Similarly, any reference that describes the use of SL-501 for Brewer QA? Otherwise some more information should be added.

151 The paper is about UV measurements. The discussion here and Figure 5 are for total ozone. I suggest to remove both.

151-157: The stability of the UV measurements should be mainly assessed by the 1kW calibration lamps which are not mentioned here, and the 50 W lamps are mainly supporting the assessment of shorter term variations.

183: Most spectral quantities extend beyond the spectral range of the measurements. How is this taken into account and what are the uncertainties involved?

185-195: The list of different quantities could be supported by a figure with action spectra and a description of how these effective does are calculated. Of course all these are already published elsewhere, but for the completeness of the presentation it would be useful to be included.

220-221: Again the paper is focused on UV measurements. It could discuss briefly ozone, but showing a figure for ozone (Figure 8) is too much. I suggest removing this figure.

369: The quality of the flow chart could be improved

386: The quality of the flow chart is very poor.

[Figure]

42, 2016.

Interactive
comment

---

## Author Comment (AC1) · 27 Apr 2016

Authors' response to the review of the manuscript "Data flow of spectral UV measurements at Sodankylä and Jokioinen" by Mäkelä et al.

The Authors appreciate the constructive comments of the Referee #1 and respond here below to each remark. The comments/questions presented by the referee are indicated as C. The answers are indicated as A. The manuscript has been upgraded following the referee's comments. The corresponding changes in the manuscript are indicated as U.

The revised manuscript has been uploaded as supplement.

Major comments

[Figure]

C: The English language used throughout the paper needs major improvements. There are various cases that words are missing or sentences are difficult to understand.

A: The authors agree with the referee on this point.

U: The language has been improved and corrected to fulfill the requirements of a scientific publication.

C: Introduction, Paragraph 1 "The Brewer was designed to measure ozone but then UV and now SO2 is measured". The correct way to write that, is that the instrument was initially designed to measured total column ozone with the differential absorption method (reference) using the direct sun port. In addition, the global (diffuser) port was introduced for measurements of spectral UV. Then the use of the direct sun data have used in order to calculate SO2 (reference), Aerosol Optical depth (e.g. Groebner, Kazadzis, Marenco ..), NO2 (e.g. Cede, Diemoz,..)

A: We fully agree. The sentences have been modified as suggested.

U: The text now reads: The Brewer spectrophotometer (Brewer) was initially designed to measure total column ozone with the differential absorption method (Bais et al. 1996; Brewer, 1973) using the direct sun port. In addition, a global (diffuser) port was introduced for measurements of spectral UV. The direct sun data have also been used to calculate SO2 (Cappellani and Bielli, 1995), aerosol optical depth (Gröbner et al. 2001; Kazadzis et al. 2005; Marenco et al. 2002) and NO2 (e.g. Cede et al. 2006; Diémoz et al. 2014 ).

C: Paragraph 2 It is interesting to describe this dynamic range and in addition some details about the difficulty for such measurements for places like e.g. Sodankyla where low solar eleva- tions is challenging for UV measurements.

A: In deed, this is an interesting issue. Discussion on the magnitude of the dynamical range and the effect of high SZA is now included.

U: The following text has been added: "The high dynamical range of five to six orders of magnitude of UV irradiance reaching the surface of the Earth puts great demands to the instruments designed to monitor both the short UV-B wavelengths (290-315 nm) and the longer UV-A wavelengths (315-400 nm). The challenge is to maintain the sensitivity of the instrument at all wavelengths. And: "The location of Finland at high latitude, where high solar zenith angles (SZA) are frequent, brings additional challenges, as the weak signal at UV-B wavelengths is near the noise level of the instrument."

C: Paragraph 3 -It is interesting to mention a recent intercomparison campaign. But these results are not yet published. So in addition to this you could add results of a large number of pre- vious publications that report such differences. (for example the SUSPEN paper by A. Bais and various others also in form of EU publications related with the QASUME trav- elling standard). In addition, describing this recent campaign someone has to clarify if this 20% down to 6% calibration differences are due to: primary calibration sources dif- ferences and processing differences or (/and) changes in the instruments due to their travel to the campaign site.

A: The text has been modified according to the suggestion from the referee. The text regarding the use of a common lamp as calibration source has been excluded, as recent findings have given reasons to be careful in making conclusions based on the measurements done with the common lamp.

U: The following text has been added: "The difficulty of the absolute calibration was already seen in the intercomparison campaigns of the 1990's (Josefsson et al. 1994, Koskela et al. 1997) and in the twenty-first century (e.g., Bais et al. 2001), in which the range of the deviations from the reference for UV spectra was up to ±20%. Despite the efforts to homogenize the measurements, in the last European Brewer comparison organized by the COST 1207 project in El Arenosillo, Spain, six Brewers out of 18 differed by more than 10% from the reference, when using the calibration provided by the operator (http://www.pmodwrc.ch/wcc_uv/wcc_uv.php?topic=qasume_audit). The differences are most likely due to slightly different corrections (for, e.g, temperature dependence and angular response) and processing procedures. Small variations in a number of corrections and procedures may result in large differences in the outcome."

C: Last paragraph of the introduction -You are mentioning steps like: acquisition, processing, storage, and dissemination. I think that a proper way to describe all these steps is to start from : a. Acquisition and Quality control of raw or e.g. level 0 data. b. Calibration of the measured data c. Quality control and corrections (spikes, straylight, dead time, etc) towards e.g. level 1 data. d. Online real time checks e. Post processing of spectral data and creation of e.g. level 2 or metadata

A: The structure of the paper has been changed following the suggestion to clearly show different levels of data.

U: The structure is now: 3 Data flow 3.1 UV data acquisition 3.2 IDEAS - A quality control tool 3.3 UV data processing 3.3.1 Calibration with Level 1 and Level 2 responsivities 3.3.2 Processing algorithms 3.3.3 Online processing - Level 1 data 3.3.4 Offline processing - Level 2 data 3.3.5 Products

And the following explanatory paragraph has been added: "This study examines and demonstrates the steps that are involved in processing different levels of solar spectral UV irradiance data produced by the Brewer spectrophotometers in possession of the Finnish Meteorological Institute. Due to economical reasons, the Brewer measurements at Jokioinen were terminated in November 2015. Since then, the Brewer #107 has been relocated and operated in Helsinki. Thus, this study also serves as a historical description of the Jokioinen measurements, and as a platform for the development of the procedures to be followed in Helsinki and Sodankylä in the future. A detailed description is given on the process flow from the Level 0 data (raw counts) to the Level 2 data (quality assured spectral UV irradiances and UV products). In a companion paper (Mäkelä et al. 2016) we describe in detail how the final time series of the responsivity of a Brewer spectrophotometer is derived. In another companion paper (Heikkilä et al. 2016) we describe how the quality indicators provided by the European UV database (EUVDB) may be used for quality assurance of Level 2 data."

C: Section 2 It would be useful to provide some graphical example of the data availability from the beginning of the measurements till today. Also a simple graph showing the calibration record (e.g. instrument responses) for all the period, accompanied with the calibration uncertainties that can be probably differ from the early 90s till today.

A: Histograms on the monthly amounts of Level 2 data submitted to the EUVDB has been added in Section 3.3.4.

U: A graph showing the time series of the responsivities of Brewer #037 and #107 has been added to Section 3.3.4 Information on the calibration uncertainty has been added to the text of section 3.3.1 as follows: "Bais et al. 2006 have calculated that the uncertainty of the UV irradiances due to the calibration uncertainty is 1.4%."

C: Section 3 In order to make all the individual steps clear, I would stick to a format similar than the one described in the end of my comments for paragraph 1. It is important to make the individual steps very clear as this is all that the paper is about.

A: The manuscript has been restructured to make the steps clear. In addition, the figures with the schematic presentations on the data flow have been restructured.

C: A table with the initial steps is essential. But more essential is to accompany this table with the references describe each of the steps. (e.g. figure 6 could be modified towards this goal)

A: Figure 6 has been modified towards to goal suggested by the referee. Figure 4 and Figure 6 have been both restructured.

U: The manuscript now includes a chart giving an overall view on the whole data flow and the steps required to produce the different levels of data, and another chart on the online (NRT) processing scheme.

C: Figure 5 . Instead of providing a print screen image here. You can show internal lamp time series and examples of good and bad measurements and Corrections.

A: As this manuscript includes only UV data flow, and not the ozone data flow, the Figure has been removed as suggested by the Referee #2. The internal lamp results are not directly used for the quality control of UV measurements. Of course if some dramatic change is seen in the internal lamp time series, there is a doubt that something has happened to the instrument, which also affect the UV measurements, and the instrument need to be recalibrated.

C: It is not clear if the IDEAS software is applied in the raw data or in the calibrate & corrected ones. I think you have to report that the software assures of specific "bad data" or instrument failures and not an evaluation of the post correction or calibration methods that are applied.

A: As the manuscript is restructured, the section discussing IDEAS is now placed between sections dealing with "UV data acquisition" and "UV data processing", to clearly indicate the purpose of IDEAS to act as a QC tool for the Level 0 data.

C: Instead of screenshots of "possible warnings" for describing the QA/QC you have to report on all warnings that are included in the process.

A: The description of IDEAS was indeed not as clear as it could be. Among other processes the system identifies the internal test failures that potentially lead to poor data quality, which is now clearly stated in the text. Examples of critical tests are now provided. Well over a hundred of warning types are included in the process and the list is growing as IDEAS is actively being developed. While it would be interesting to include all warning types that are provided by the system, we believe that the list is too long to be part of this paper.

C: I do not understand why "if necessary a wavelength correction is made." How do you know that is necessary or not if you do not already have a report on possible wavelength shifts ?

A: The wavelength correction is done to all spectra. It is based on the shape of the extraterrestrial spectrum.

U: The text has been changed to: "The SHICRIVM algorithm (Slaper et al 1995) is used to correct for wavelength shifts."

C: Personal remark Copying from the manuscript: "These instruments form an important network for monitoring changes in the total ozone column. Nowadays, these spectral UV time series of over twenty years are unique and among the longest measured in the Arctic. " and "The observatory at Jokioinen is in the process of being shut down, and the spectral UV measurements have been moved to Helsinki. Thus, this paper also serves as a historical description of the Jokioinen measurements."

So Jokioinen is a "unique" and for sure among the longest not only in the arctic but worldwide, station for ozone and UV measurements but it is in the process of being shut down. Since this paper is co-authored and co-signed from 12 FMI scientists, I want to pose the question: how it is possible to refer it here as a unique and historical station and at the same time to shut it down? And of course moving the instruments somewhere else is not an ideal way of a continuation of this 20 year series.

So for me it seems at least ironic to say that this paper is important (also) because Jokioinen is a unique station in terms of time series. Now, a paper that would show a full 20 year time series of Jokioinen Brewer measurements, calibration efforts, e.t.c. sure would serve as a historical description. On the other hand describing procedures for online quality control (only recently added) for a station that closes down are not useful for a future user of the data, as much as a report/publication on the actual calibrations, correction procedure, uncertainty analysis and measurement trend analysis for the particular station.

A: We agree with the referee.

U: The text has been rephrased to read as follows: "Due to economical reasons, solar spectral UV measurements at the Jokioinen observatory were terminated in

November 2015, and the Brewer #107 was transferred to and set up for measurements in Helsinki, the capital city of Finland. The twenty-year time series of solar spectral UV irradiance obtained in Jokioinen is in the process of being analysed. The methods and procedures reported in this study are still in continuous use in Helsinki and Sodankylä. Brewer #037 is still operated in Sodankylä and continues to collect its over 25-year record of solar spectral UV measurements. In Helsinki, a new time series has been initiated with Brewer #107. Thorough understanding on the different phases in the data flow and further development of the methods is a pre-requisite for the vitality of the monitoring programs also in the future."

Please also note the supplement to this comment:
http://www.geosci-instrum-method-data-syst-discuss.net/gi-2015-42/gi-2015-42-AC1-supplement.pdf

**Supplement:**

**Data flow of spectral UV measurements at Sodankylä and Jokioinen**

**J.S. Mäkelä[1*], K. Lakkala[1,2], T. Koskela[1**], T. Karppinen[2], J.M. Karhu[2] , V. Savastiouk[4], H. Suokanerva[2], J. Kaurola[1], A. Arola[3], A.V. Lindfors[1], O. Meinander[1], G. Leeuw[1,5] and A. Heikkilä[1]**

[1] {Finnish Meteorological Institute, Research and Development, 00101 Helsinki, Finland}

[2] {Finnish Meteorological Institute, Sodankylä, Finland}

[3] {Finnish Meteorological Institute, Kuopio, Finland}

[4] {IOS Inc, Toronto, Canada}

[5] {Department of Physics, University of Helsinki, 00014 Helsinki Finland}

[*]{now at University of Jyväskylä, Jyväskylä, Finland}

[**]{now an independent researcher}

Correspondence to: K. Lakkala (Kaisa.Lakkala@fmi.fi)

**Abstract**

The data flow involved in a long-term continuous solar spectral UV irradiance monitoring program is investigated and structured to provide an overall view on the multi-phase process from data acquisition to the final products. The program employing Brewer spectrophotometers as measuring instruments is maintained by the Finnish Meteorological Institute (FMI) ever since the 1990's at two sites in Finland, Sodankylä (67°N) and Jokioinen (61°N). It is built upon rigorous operation routines, processing procedures, and tools for quality control (QC) and quality analysis (QA) under continuous development and evaluation. Three distinct levels of data emerge, each after certain phase in the data flow: Level 0 denoting raw data, Level 1 meaning calibrated data processed near-real-time, and Level 2 comprising of post-processed data corrected for all distinguishable errors and known inaccuracies. The final products disseminated to the users are demonstrated to result from a process with a multitude of separate steps, each required in the production of high quality data on solar UV data radiation at the Earth's surface.

1  **Introduction**

The Brewer spectrophotometer (Brewer) was initially designed to measure total column ozone with the differential absorption method (Bais et al. 1996; Brewer, 1973) using the direct sun port. In addition, a global (diffuser) port was introduced for measurements of spectral UV. The direct sun data have also been used to calculate $SO_2$ (Cappellani and Bielli,

1995), aerosol optical depth (Gröbner et al. 2001; Kazadzis et al. 2005; Marenco et al. 2002)

and $NO_2$ (e.g. Cede et al. 2006; Diémoz et al. 2014 ). At present, there are over 220

instruments set up by research institutes all over the world (http://kippzonen-brewer.com/).

These instruments form an important network for monitoring changes in the atmospheric total ozone column and solar UV irradiance at the Earth's surface.

The first Brewer spectrophotometers in Finland were set up in 1990 and 1995, in Sodankylä

and Jokioinen, respectively, to respond to the need to monitor total ozone and UV radiation after the discovery of the Arctic ozone loss. These time series of solar spectral UV irradiance of over twenty years are unique and among the longest measured in the Arctic. The homogenized time series have been used for several studies related to Arctic ozone loss (e.g.,

Bernhard et al. 2013; Manney et al. 2011; Knudsen et al. 1998), and validation of satellite data (Hassinen et al. 2008). They have also been used in, e.g., studies on the effects of UV

radiation on biological objects (e.g., Lappalainen et al. 2010; Martz et al. 2009), materials (Heikkilä 2014) and human health (Kazantzidis et al. 2009).

The Brewer spectrophotometer is a versatile but also a complex instrument in comparison to, for instance, broadband UV meters. Many intermediate steps and corrections are needed in the processing of the data. The high dynamical range of five to six orders of magnitude of UV

irradiance reaching the surface of the Earth puts great demands to the instruments designed to monitor both the short UV-B wavelengths (290-315 nm) and the longer UV-A wavelengths (315-400 nm). The challenge is to maintain the sensitivity of the instrument at all wavelengths. The location of Finland at high latitude, where high solar zenith angles (SZA)

are frequent, brings additional challenges, as the weak signal at UV-B wavelengths is near the noise level of the instrument. High quality data can only be ensured after careful characterization of the instrument, correction of known measurement errors and careful quality control (QC) and quality assurance (QA). (Seckmeyer et al. 2001; Garane et al. 2006; Lakkala et al. 2008; Webb et al. 2003).

Maintaining a Brewer spectrophotometer absolutely calibrated is a demanding task (Bernhard and Seckmeyer 1999, Webb et al. 1998). International campaigns are regularly organized to evaluate the calibration and measurement procedures performed by different Brewers and institutes. The difficulty of the absolute calibration was already seen in the intercomparison campaigns of the 1990's (Josefsson et al. 1994, Koskela et al. 1997) and in the twenty-first century (e.g., Bais et al. 2001), in which the range of the deviations from the reference for UV spectra was up to ±20%. Despite the efforts to homogenize the measurements, in the last European Brewer comparison organized by the COST 1207 project in El Arenosillo, Spain, six Brewers out of 18 differed by more than 10% from the reference, when using the calibration provided by the operator (http://www.pmodwrc.ch/wcc_uv/wcc_uv.php?topic=qasume_audit). The differences are most likely due to slightly different corrections (for, e.g, temperature dependence and angular response) and processing procedures. Small variations in a number of corrections and procedures may result in large differences in the outcome.

For the comparability of the Brewer data from around the world, it is necessary to carefully document the traceability of the calibration and the processing chain of the data. Rigorous documentation should be a part of the routine QC/QA procedures at each site. This allows anyone to audit all the steps taken prior to the delivery of the data, and allows making changes in post-processing with no need to start everything from the beginning.

This study examines and demonstrates the steps that are involved in processing different levels of solar spectral UV irradiance data produced by the Brewer spectrophotometers in possession of the Finnish Meteorological Institute. Due to economical reasons, the Brewer measurements at Jokioinen were terminated in November 2015. Since then, the Brewer #107 has been relocated and operated in Helsinki. Thus, this study also serves as a historical description of the Jokioinen measurements, and as a platform for the development of the procedures to be followed in Helsinki and Sodankylä in the future. A detailed description is given on the process flow from the Level 0 data (raw counts) to the Level 2 data (quality assured spectral UV irradiances and UV products). In a companion paper (Mäkelä et al. 2016) we describe in detail how the final time series of the responsivity of a Brewer spectrophotometer is derived. In another companion paper (Heikkilä et al. 2016) we describe how the quality indicators provided by the European UV database (EUVDB) may be used for quality assurance of Level 2 data.

**2    Description of the stations**

The Finnish Meteorological Institute (FMI) has been operating Brewer spectrophotometers at two sites: Sodankylä and Jokioinen. In the following sections, brief descriptions on the characteristics of the sites and their Brewer spectrophotometers are given.

**2.1    The Sodankylä station (Brewer #037 and #214)**

The Arctic Research Centre of the FMI is located at 67.37°N, 27.63°E, at 179 m of altitude above sea level, in Sodankylä. It has maintained Brewer Mark II single monochromator spectrophotometer since 1990. The nearby surroundings comprises of pine forests. In the southwest flows the river Kitinen. The area is surrounded by a large peatland area in the east. The terrain is snow covered typically from October to late April. The sun is just below the horizon from mid-December to mid-January. Temperatures range from -40 °C in winter to +30 °C in summer. The station is described in more detail in, e.g., Lakkala et al 2003.

Two Brewers are currently operated at this site: #037 and #214. They are located on the roof of the sounding station (see Fig. 1). Brewer #037 has a single monochromator and a wavelength range of 290-325 nm. The FWHM of the slit function is 0.56 nm. The other Brewer, #214, has been set up in 2012, to work in tandem with Brewer #037 and to cover the longer wavelengths of the UV radiation. The wavelength range of Brewer #214 is 286.5-365 nm. The FWHM of the slit function of Brewer #214 is 0.55 nm.

**2.2    The Jokioinen station (Brewer #107)**

The meteorological observatory in Jokioinen is located at 60.82°N, 23.50°E, at 107 m of altitude above sea level, in a rural area surrounded by fields and mainly coniferous forests. The ground is covered by snow most of the time during December-March. Temperatures can range from -20 °C to +30 °C.

FMI acquired and set up Brewer #107 in Jokioinen in 1995. Since November 2015, it has been relocated in Helsinki. At Jokioinen, Brewer #107 was located on the roof of the sounding station (see Fig. 2). Brewer #107 is of type Mark III with a double monochromator. The original wavelength range of the instrument was 286.5-363 nm. In April 1997, changes in the optics were made. Since then, the wavelength range has been 286.5-365 nm. The FWHM of the slit function is 0.59 nm.

**3    Data flow**

A schematic presentation on the data flow from the raw measurements performed by the FMI Brewer spectrophotometers to the final UV products is given in Fig. 3. Three levels of data are produced in the process. Level 0 data results from data acquisition, Level 1 data from the near real time (NRT) processing of Level 0 data and the following quality control (QC), and Level 2 data from post-processing of Level 0 data and the following quality assurance (QA).

To demonstrate the different phases in the data flow and the way the data is transformed as they go through the whole chain from the raw counts to final products, we use one selected case spectrum measured by Brewer #107 in Jokioinen on the 20[th] of May 2007. Information related to the case scan is given in Table 1.

**3.1    UV data acquisition – Level 0 data**

The measurements performed by Brewer spectrophotometers are controlled through their own operating computers. In UV irradiance measurements, the quartz dome viewport is used to collect the photons from the hemisphere above. The wavelength range from the shortest wavelength to the upper limit of the range is scanned in ascending direction with 0.5 nm increments. The monochromator is used to select the photons on one narrow wavelength band at a time and passed on to the photomultiplier tube (PMT) acting as a detector. The PMT makes observations in cycles: four cycles at each wavelength below 300 nm and two cycles at wavelengths above 300 nm. The raw counts are stored into files onto the operating computer in units counts/cycle. These data are the Level 0 data produced by the Brewer spectrophotometer. An example on the Level 0 data raw counts is shown in Fig. 4.

The operating computers of the Brewers work autonomously, making measurements based on schedules predefined by the operator. Over the years, a variety of schedules has been used, and the schedules for the two sites have supported slightly different research targets at different times. For example, at Jokioinen, more frequent measurements have been made near sunrise and sunset, and at constant air masses, as well as at time of the smallest solar zenith angle (SZA). At Sodankylä, measurements have generally been spread out more evenly throughout the day. At both sites, there has been a measurement at least every half hour during daylight and a measurement at midday. The number of daily UV scans at Jokioinen, for example, range from about 8 in winter to more than 30 in summer.

Other data acquired at the measurement sites and relevant to the UV data processing include visibility and total ozone column. Automatic weather stations (AWS) produces data on visibility every ten minutes. Total ozone column measurements are performed by the Brewer spectrophotometers themselves. The data acquired on visibility and total ozone column are used in the cosine correction procedure applied to the measured UV irradiance spectra (Lakkala et al. 2008). As auxiliary measurements, a broadband UVB radiometer (Solar Light SL501A) and a pyranometer (Kipp & Zonen CM11) measuring global radiation (305–2800 nm), are synchronized to the Brewer measurements. These measurements are used in offline quality assurance (QA) procedures to identify erroneous measurements, and to obtain information on changes in the cloud cover (Lakkala et al. 2008).

The Level 0 data on solar spectral UV irradiance are transferred from the Brewer's operating computer to a central UNIX server. The raw data are further transferred to another server for the quality control (QC) monitoring the proper functioning of the instrument. The system taking care of the OC is called IDEAS (Integrated Data Evaluation & Analysis System) and is briefly described in the following section. In addition, the raw data are uploaded every 20 minutes to the European Brewer Network Database (EUBREWNET, http://rbcce.aemet.es/eubrewnet). The EUBREWNET is established within the COST 1207 project of the European Union, as a joint effort of the international Brewer community.

**3.2   IDEAS – a quality control tool**

In 2015, new software was introduced to facilitate both immediate and long-term quality control of data, and to improve the potential of the Brewers to work as real-time operational devices. IDEAS (Integrated Data Evaluation & Analysis System, supplied by Full Spectrum

Science Inc.) is a tool using the Level 0 data directly for checking that the Brewer is functioning correctly. The Brewer itself makes several check measurements during the day.

The measured parameters are used to monitor the stability of the instrument.

Every measurement of the Brewer and every process of the operating software is recorded in appropriate data files. These are stored as so-called B-files, and updated to the server in which

IDEAS is running. IDEAS is integrated with the real-time 7/24 operational control system of

FMI. Automatic warnings of malfunctions of the Brewer are generated within 5-10 minutes and sent to the 24/7 control center. If the personnel there are unable to solve the problems, stand-by Brewer specialists can be alerted by text messages when needed.

The IDEAS software analyzes all records collected by the Brewer spectrophotometer operational program (see Savastiouk 2011 for a description of an earlier version of the software).  These records include observations, tests and user comments.  For each record type that is known to the system, the results are compared to nominal or reference values, prioritized and reported to the main FMI observation database system with flags indicating the state of the instrument.  For record types that are not yet known to the system a report is created so new record types can be added to the system analysis.  All data is stored in a database for easy access at any time.

The full list of currently analyzed records contains over a hundred items. Some of the most relevant critical tests include the wavelength calibration internal mercury lamp test (HG), the linearity, or dead time, test (DT) and the spectral sensitivity stability test on the internal halogen lamp (SL). Failures in any of these tests suggest that other data collected at the same time are questionable and need attention from the operator and/or a scientist. In such cases, an immediate alert is sent to the people responsible for the instrument to address these failures.

Both real-time and historical reporting tool is also part of IDEAS. The tool generates a collection of prioritized tables and plots that include test results as well as ozone data comparison with satellites as an informational aid for the user to evaluate the data quality.

**3.3 UV data processing**

For the processing of solar spectral UV irradiance data produced by the Brewer spectrophotometers, two separate lines have been established. The first is used for online near-real-time (NRT) processing, using the Level 0 data and auxiliary data as inputs to produce Level 1 data. The second is employed in offline post-processing, using the same data as inputs but employing a re-evaluated time series of responsivity of the instrument, correction for potential shifts in the wavelength scale, and final QA procedures. In both processing schemes, knowledge on the responsivity of the instrument is needed for the production of calibrated irradiance data.

**3.3.1 Calibration with Level 1 and Level 2 responsivities**

In order to convert the measured photon counts into irradiances, the response of the instrument must be known. The response is determined using 1 kW lamp measurements performed in an optical laboratory. The optical laboratory facilities at Sodankylä are described in Lakkala et al. 2016. The optical laboratory facilities at Jokioinen are similar to those at Sodankylä.

The calibration lamps are tungsten-filament incandescent halogen lamps of type DXW

operated in vertical beam direction. A primary standard is used to transfer the irradiance scale from the National Standard Laboratory MIKES-Aalto. Using the measurements of the

Brewer, the scale is transferred to working standards used for the calibration of the Brewer every six weeks. Usually at least three lamps are measured during one calibration event, which enables the detection of potential drifts in the lamps (Webb et al. 1998, Lakkala et al.

2008). The obtained responsivity is used in the online (NRT) processing and production of

Level 1 data until the next calibration. Garane et al. 2006 have calculated that the typical uncertainty of the UV irradiances due to the absolute calibration uncertainty is 1.4%.

The determination of the Level 2 responsivity with daily values is briefly described in

Lakkala et al. (2008) and is covered in more detail in the companion paper (Mäkelä et al.

2016). In brief, calibration lamp measurements are analysed back in time on a longer time span and measurements of all lamps are incorporated in cross-checking and comparison, to separate the long-term drifts of the lamps from transient exceptions in their output, and to confirm the selection of the core "trusted" lamps that serve as the basis for the final Level 2

calibration. This analysis results in a response time series with daily responsivities over the time period investigated. The time series is formed by linear interpolation between the moments of lamp measurements and smoothed by using a moving average.

Examples on the responsivities are shown in Fig. 4. The Level 1 and Level 2 responsivities plotted in the figure have been applied in the online (NRT) and offline post-processing of the raw counts recorded by Brewer #107 on 20 May 2007 (the case scan described in Table 1).

The interpolated and smoothed time series of Level 2 responses at 305 nm for Brewer #037 is shown in Fig. 5a to demonstrate the temporal development of the response of the instrument.

In Fig. 5b, the stepwise time series of Level 1 and the interpolated and smoothed time series of Level 2 responsivities at 305 nm for Brewer #107 are also shown partly overlapping to illustrate the difference between the responsivities used in Level 1 and Level 2 processing.

Between the 1 kW lamp calibrations performed in the laboratory, the stability of the Brewer is monitored on-site every three weeks using external 50 W lamp measurements.  In those measurements, at least three lamps are used as well. If a change in the spectral response of the

Brewer is detected, the Brewer is moved to the laboratory for absolute calibration.

**3.3.2  Processing algorithms**

Both the online and offline data processing are done using custom-made software written in
Perl (Practical Extraction and Report Language) and supported by shell scripts. The original
software provided by the instrument manufacturer and the algorithms therein has served as a
basis for the software development. The algorithms have been described in detail by Lakkala
et al. (2008). Only the key points are therefore summarized in the following.

The Brewers perform the scans by starting from the shortest and ending at the longest
wavelength of the wavelength range of the instrument. The Brewer software also returns the
dark current, F1, for each scan. The total scanning time is about 4 minutes for Brewer #037
and 5 minutes for Brewers #107 and #214 due to the larger wavelength range. The
occasionally occurring noise spikes are first removed based on the method of Meinander et al.
(2003). Following SCI-TEC Instruments Inc. (1999), the raw counts F (in units counts/cycle)
recorded by the Brewer are next converted to count rates C (counts/second) by subtracting the
dark counts F1 and taking into account the integration time IT and the number of cycles CY
of the slit exposure. The integration time is pre-defined as 0.1147 s. For each wavelength i,
where i=290-325(365)nm, the dark current corrected count rate is calculated as

$$C_i = \frac{2(F_i - F1)}{CY * IT} \tag{1}$$

The dead time DT of the photomultiplier is measured daily, and taken into account using
Poisson statistics. Nine iteration rounds (n=0-8) are next used to correct the count rate for the
dead time:

$$C_i(n) = C_i(0) * e^{C_i(n)*DT} \tag{2}$$

The stray light is calculated as the average of all counts below 292 nm (#107 and #214) or
293 nm (#037). The count rate corrected for stray light is simply

$$C_i = C_i - \frac{1}{N} \sum_{i<292(3)} C_i \qquad\qquad (3)$$

where N is the number of wavelengths below the stray light limit. Since #107 and #214 have a double monochromator, the stray light counts are small. The Sodankylä Brewer #037 has a single monochromator, and the stray light counts are larger (Bais et al. 1996).

The count rates are then converted to irradiances by dividing the count rate $C_i$ with the spectral response $R_i$ of the instrument:

$$I_i = \frac{C_i}{R_i} \qquad\qquad (4)$$

Temperature and cosine corrections are then made to the spectral irradiances. A linear depencence for temperature is assumed:

$$I_i = I_i * (1 + CT_i * \Delta T), \qquad\qquad (5)$$

where $CT_i$ is the temperature correction coefficient and $\Delta T$ is the temperature difference between the measurement temperature and the reference temperature.

The cosine correction is based on radiative transfer calculations (Lakkala et al. 2008). As both the direct and the diffuse component of the global radiation affect the cosine correction factor, we need to estimate the ratio of the direct and diffuse components of the actual global radiation. This is done using the freely available libRadtran package (Mayer and Kylling 2005). The only unknown input parameter to the model is the cloud optical depth, which is estimated using a predefined lookup table. The inputs to the lookup table include the total ozone (measured with the Brewer), the irradiance given by the Brewer, the solar zenith angle, the visibility (given by the AWS), and the albedo. Currently, a fixed albedo value of 0.03 is used for both stations, corresponding to the average summer albedo.

**3.3.3 Online processing - Level 1 data**

The components of the online NRT processing scheme are described in Fig. 6. Processing of all the data from both stations is done at a single central Unix server. The raw (Level 0) data is transferred from the Brewer operational computer to the central server every five minutes.
From the central computer the data is further transferred to the IDEAS server, where the QC
of the data is done. Every five to ten minutes status messages are sent to the observation
control server, from which the critical status messages are transferred to the 24/7 control of
FMI. Level 0 data is also transferred to the European Brewer database, EUBREWNET every
20 minutes. As soon as a new UV measurement has been detected by the central server, the
data processing of Level 1 data starts and visibility is downloaded from the climate database
of FMI, and used for the cosine correction of the data. The Level 1 data is then used to
calculate products which are transferred to a server with a web interface and the climate data
base of FMI. The reference data from the broadband SL501A radiometer and the global
radiation pyranometer CM11 are downloaded each time a spectral measurement is made. An
example on the Level 1 irradiance spectrum is given in Fig. 7.

The Level 1 data is spectral UV irradiance processed in near real time. Several corrections are
routinely made in the Level 1 processing, and several data products are derived. In addition to
the near-real-time measurements, every morning the data from the previous day is
reprocessed. This is mainly done to avoid data gaps in Level 1 data. Such gaps might for
example occur if there has been a malfunction of the central server, in which case the Level 1
spectra have not been calculated even though the Level 0 data from the Brewer are still
available.

### 3.3.4   Offline processing - Level 2 data

Level 1 spectral irradiance data is produced using the Level 1 responsivity updated regularly
on the basis of the calibration lamp measurements. To produce Level 2 data, calibration lamp
measurements are analysed back in time to produce response time series as described in
Section 3.3.1. By this procedure, the potential drifts in the calibration lamps, possibly
affecting the Level 1 responsivity, are eliminated. This correction can be calculated once a
year, when the primary calibration lamps are recalibrated.  The Level 2 data is fully
homogenized and quality assured. The SHICRIVM algorithm (Slaper et al 1995) is used to
correct for wavelength shifts. As a quality control tool, the dose rates are compared with reconstructed UV, model calculations of clear sky UV and global radiation, global radiation and broad band UV data in order to distinguish erroneous measurements (Lakkala et al. 2008). In addition each spectrum is checked by eye and bad measurements are excluded. An example on the Level 2 irradiance spectrum is given in Fig. 7.

Once a year the Level 2 data is uploaded to the database EUVDB (http://uv.fmi.fi/uvdb/). Figure 8 shows the number of UV spectra submitted to the EUVDB measured by the Brewer #037 (Fig. 8a) and Brewer #107 (Fig. 8b) during 1990-2014 and 1995-2014, respectively. The submission of  Brewer #037 spectra for 2011 will be upgraded in near future, as there exists more than 5700 spectra for that year, but only some of them have been submitted to the database at the time of writing the manuscript. The data from Sodankylä is also uploaded to the database of the FMI-Arctic Research Centre (http://litdb.fmi.fi/).

**3.3.5   Products**

Several products are derived from the Level 1 and Level 2 spectral data. These include dose rates (in W m$^{-2}$) and daily doses (in J m$^{-2}$) as unweighted and weighted by selected action spectra (Table 2 and Fig. 9). Let us denote an arbitrary action spectrum by $s(\lambda)$. The dose rate $\dot{D}$ is calculated from the irradiance spectrum, multiplied by the action spectrum, by numerical integration over the appropriate wavelength range (UV, UVA or UVB):

$$\dot{D} = \int_{\lambda_{min}}^{\lambda_{max}} s(\lambda) \cdot E(\lambda) d\lambda. \qquad (6)$$

For the unweighted quantities, $s(\lambda)$ is equal to unity. The daily dose is further derived from the dose rate by numerical integration over the day:

$$D = \int_{t_{min}}^{t_{max}} \dot{D} \, dt. \qquad (7)$$

The derivation of the products is illustrated in Fig. 10. The action spectra currently in routine use are listed in Table 2. UV index is additionally derived by multiplying the CIE erythemally weighted UV dose rate by 40 (WMO 1997).

For the calculation of the dose rates requiring integration beyond the upper wavelength limit of the Brewer, the measured spectra are extended using a pre-defined reference UVA spectrum (Fig. 10). The extension is adjusted onto the level of the measured spectrum by linear conversion. The ratio of the measured irradiance to the reference irradiance at selected wavelength is used as a scaling factor. For Brewer #037, the wavelength of 324 nm, and for Brewer #107 and #214, the wavelength of 361 nm is used as a point of adjustment. All action spectra in routine processing (Fig. 9) approach zero towards the longer UVA wavelengths. This means that the uncertainty caused by the artificial UVA extension to the computed dose rate is of the order of $10^{-3}$. For the unweighted UV and UVA dose rates, our investigation based on a radiative transfer model simulation suggest uncertainties as high as approx. 2 % caused by the constant scaled UVA extension. This finding is in line with the result obtained by Fioletov et al. 2004.

All products are stored in the central server, and graphs and statistical tables are updated each time a new measurement is processed. The data are transferred to a server which has a web interface for internal use at FMI. This information is used at FMI for operational and research purposes. In addition, the Level 1 UV index is transferred NRT to the FMI climate database.

**4    Conclusions**

Production of high quality data on solar spectral UV at the Earth's surface requires a complex set of operation routines, processing algorithms, and QC/QA procedures. We have described and demonstrated the methods used in the acquisition and processing of solar spectral UV irradiance data measured by Brewer spectrophotometers in Finland. As a result, we have produced a comprehensive view on the multi-phase data flow from the collection of single UV photons to calculation of final products of UV irradiance. This is expected to facilitate identification and isolation of specific targets of development in the procedures used in different phases of the data flow. The first targets of this kind appear to be the implementation of solar zenith angle and total column ozone dependent UVA extension to the measured spectra in the calculation of the different UV dose rates. Another emerging idea has been development of a comprehensive data and metadata management system to support the data flow as a whole and to ensure its continuance.

Due to economical reasons, solar spectral UV measurements at the Jokioinen observatory were terminated in November 2015, and the Brewer #107 was transferred to and set up for measurements in Helsinki, the capital city of Finland. The twenty-year time series of solar spectral UV irradiance obtained in Jokioinen is in the process of being analysed. The methods and procedures reported in this study are still in continuous use in Helsinki and Sodankylä. Brewer #037 is still operated in Sodankylä and continues to collect its over 25-year record of solar spectral UV measurements. In Helsinki, a new time series has been initiated with Brewer #107. Thorough understanding on the different phases in the data flow and further development of the methods is a pre-requisite for the vitality of the monitoring programs also in the future.

**Acknowledgements**

Professor Esko Kyrö is acknowledged for starting the Brewer measurements at Sodankylä. We are grateful to the operators at Sodankylä and Jokioinen stations for daily maintenance and for performing the calibrations of the Brewers. We thank the Brewer community within the COST 1207 project for sharing expertise related to Brewer measurements.

**References**

Bais, A. F., Gardiner, B. G., Slaper, H., Blumthaler, M., Bernhard, G., McKenzie, R., Webb, A. R., Seckmeyer, G., Kjeldstad, B., Koskela, T., Kirsch, P. J., Gröbner, J., Kerr, J. B., Kazadzis, S., Leszczynski, K., Wardle, D., Josefsson, W., Brogniez, C., Gillotay, D., Reinen, H., Weihs, P., Svenoe, T., Eriksen, P., Kuik, F. and Redondas, A.: SUSPEN intercomparison of ultraviolet spectroradiometers, J. Geophys. Res., 106(D12), 12509–12525, doi:10.1029/2000JD900561.

Bais, A., Zerefos, C. and McElroy, C.: Solar UVB measurements with the double- and single-monochromator Brewer Ozone Spectrophotometers, Geophys. Res. Lett., 23, 833–836, 1996.

Bernhard, G., Dahlback, A., Fioletov, V., Heikkilä, A., Johnsen, B., Koskela, T., Lakkala, K. and Svendby, T.: High levels of ultraviolet radiation observed by ground-based instruments below the 2011 Arctic ozone hole, Atmos. Chem. Phys., 13, 10573-10590, www.atmos-chem-phys.net/13/10573/2013/ doi:10.5194/acp-13-10573-2013, 2013.

Bernhard, G. and Seckmeyer, G.: Uncertainty of measurements of spectral solar UV
irradiance, J. Geophys. Res.,104, D12, 14,321-14,345, 1999.

Brewer, A. W.: A replacement for the Dobson spectrophotometer?. Pure Appl. Geophys.,
106-108, 919–927, 1973.

Caldwell, M. M., Camp, L.B., Warner, C. W. and Flint, S.D.:Action spectra and their key role
in assessing biological consequences of solar UV-B radiation change. In: Worrest, R.C. and
M.M. Caldwell (Eds.): Stratospheric ozone reduction, solar ultraviolet radiation and plant life.
Springer-Verlag, Berlin. 87-111, 1986.

Cappellani, F. and Bielli, A.: Correlation between $SO_2$ and $NO_2$ measured in an atmospheric
column by a Brewer spectrophotometer and at ground-level by photochemical techniques,
Environmental Monitoring and Assessment, Vol 35, 2, 77-84, 1995.

Cede, A., J. Herman, A. Richter, N. Krotkov, and Burrows, J.: Measurements of nitrogen
dioxide total column amounts using a Brewer double spectrophotometer in direct Sun mode,
*J. Geophys. Res. 111, D05304, doi:10.1029/2005JD006585*, 2006.

CIE (International Commission on Illumination), Action spectrum for the production of
previtamin D3 in human skin, CIE, 174, 2006.

Diémoz, H., Siani, A. M., Redondas, A., Savastiouk, V., McElroy, C. T., Navarro-Comas, M.,
and Hase, F.: Improved retrieval of nitrogen dioxide ($NO_2$) column densities by means of
MKIV Brewer spectrophotometers, Atmos. Meas. Tech., 7, 4009-4022, doi:10.5194/amt-7-
4009-2014, 2014.

Eleftheratos K., Kazadzis, S., Zerefos, C. S., Tourpali, K., Meleti, C., Balis, D., Zyrichidou,
I., Lakkala, K., Feister, U., Koskela, T., Heikkilä, A. and Karhu, J. M.: Ozone and
Spectroradiometric UV Changes in the Past 20 Years over High Latitudes, Atmosphere-
Ocean, DOI: 10.1080/07055900.2014.919897, 2014.

Fioletov, V. E., M. G. Kimlin, N. Krotkov, L. J. B. McArthur, J. B. Kerr, D. I. Wardle, J. R.
Herman, R. Meltzer, T. W. Mathews, and J. Kaurola: UV index climatology over the United
States and Canada from ground-based and satellite estimates, J. Geophys. Res., 109, D22308,
doi:10.1029/2004JD004820, 2004.

Garane, K., Bais, A. F., Kazadzis, S., Kazantzidis, A., and Meleti, C.: Monitoring of UV
spectral irradiance at Thessaloniki (1990–2005): data re-evaluation and quality control Ann.
Geophys., 24, 3215–3228, 2006.

Grobner, J., R. Vergaz, V. E. Cachorro, D. V. Henriques, K. Lamb, A. Redondas, J. M.
Vilaplana, and Rembges, D.: Intercomparison of aerosol optical depth measurements in the
UVB using Brewer spectrophotometers and a Li-Cor spectrophotometer, Geophys. Res. Let.
28, 1691-1694, 2001.

de Gruijl, F.R. and van der Leun, J. C.: Estimate of the wavelength dependency of ultraviolet
carcinogenesis in humans and its relevance to the risk assessment of a stratospheric ozone
depletion. Health Phys., vol 67, 319-325, 1994.

Hassinen S., Tamminen, J., Tanskanen, A., Koskela, T., Karhu, J. M., Lakkala, K. and
Mälkki, A.: Description and Validation of the OMI Very Fast Delivery Products, J. Geophys.
Res., 113, D16S35, doi:10.1029/2007JD008784, 2008.

Heikkilä, A., Methods for assessing degrading effects of UV radiation on materials, Finnish
Meteorological Institute Contributions, 111, ISBN 978-951-697-843-0, Unigrafia Oy,
Helsinki, 2014.

Heikkilä, A., Kaurola, J., Lakkala, K., Karhu, J.M., Kyrö, E., Koskela, T., Engelsen, O.
Slaper, H. and Seckmeyer, G., European 1 UV DataBase (EUVDB) as a repository and
quality analyzer for solar spectral UV irradiance monitored in Sodankylä, Geosci. Instrum.
Method. Data Syst. Discuss., doi:10.5194/gi-2015-39, in review, 2016.

Josefsson, W., Koskela, T., Dahlback, A. and Eriksen, P.: Spectral sky measurements. In The
Nordic intercomparison of ultraviolet and total ozone instruments at Izaña from 24 October to
5 November 1993. Final report, edited by Koskela, T., Finnish Meteorological Institute,
Meteorological publications No. 27, Helsinki, 73-80, 1994.

Kazadzis, S., Bais, A., Kouremeti, N., Gerasopoulos, E., Garane K., Blumthaler, M.,
Schallhart, B. and Cede A.: Direct spectral measurements with a Brewer spectroradiometer:
Absolute calibration and aerosol optical depth retrieval, Appl. Opt., 44(9), 1681 – 1690, 2005.

Kazantzidis, A., Bais, A., Zempila, M., Kazadzis, S., den Outer, P., Koskela, T. and Slaper, H.:
Calculations of the human Vitamin D exposure from UV spectral measurements at three
European stations, Photochem. Photobiol. Sci., 2009, 8, 45-51, 2009.

Knudsen, B., Larsen, N., Mikkelsen, I., Morcette, J.-J., Braahten, G., Kyrö, E., Fast, H.,
Gernand, H., Kanzawa, H., Nakane, H., Dorokhov, V., Yushkov, V., Hanse, G., Gil, M. and
ShearmanR.:. Ozone depletion in and below the Arctic vortex for 1997. Geophys. Res. Lett.,
25, 627-630, 1998.

Koskela, T., Johnsen, B., Bais, A., Josefsson, W. and Slaper, H., 1997: Spectral sky
measurements. In Nordic intercomparison of ultraviolet and total ozone instruments at Izaña
October 1996. Final report, edited by Kjeldstad, B., Johnsen, B. and Koskela, T., Finnish
Meteorological Institute, Meteorological publications No. 36, Helsinki, 109-148, 1997.

Lakkala, K., Kyrö, E. and Turunen, T: Spectral UV Measurements at Sodankylä during 1990–
2001, Journal of Geophysical Research, 108, D19, 4621, doi:10.1029/2002JD003300, 2003.

Lakkala, K., Arola, A., Heikkilä, A., Kaurola, J., Koskela, T., Kyrö, E., Lindfors, A. V.,
Meinander, O., Tanskanen, A., Gröbner, J. and Hülsen. G.: Quality assurance of the Brewer
spectral UV measurements in Finland, Atmos. Chem. Phys., 8, 3369–3383, 2008.

Lakkala, K., Suokanerva, H., Karhu, J.M., Aarva, A., Poikonen, A., Karppinen, T., Ahponen,
M., Hannula, H.-R., Kontu, A. and Kyrö. E.: Optical laboratory facilities at the Finnish
Meteorological Institute − Arctic Research Centre. Geosci. Instrum. Method. Data Syst.
Discuss., doi:10.5194/gi-2015-43, 2016.

Lappalainen N., Huttunen, S., Suokanerva, H. and Lakkala, K.: Seasonal acclimation of the
moss *Polytrichum juniperinum Hedw*. to natural and enhanced ultraviolet radiation, Environ
Pollut., 158, 3, 891-900, 2010.

Manney G., Santee M., Rex M., Livesey N., Pitts M., Veefkind P., Nash E., Wohltmann I.,
Lehmann R., Froidevaux L., Poole L., Schoeberl M., Haffner D., Davies J., Dorokhov V.,
Gernandt H., Johnson B., Kivi R., Kyrö E., Larsen N., Levelt P., Makshtas A., McElroy T.,
Nakajima H., Parrondo M., Tarasick D., von der Gathen P., Walker K. and Zinoviev N.:
Unprecedented Arctic ozone loss in 2011, Nature, 478, 469–475, doi:10.1038/nature10556,
2011.

Marenco, F., A. di Sarra, and De Luisi, J.:Methodology for determining aerosol optical depth
from Brewer 300-320-nm ozone measurements, Appl. Opt., 41, 1805-1814, 2002.

Martz F., Turunen, M., Julkunen-Tiitto, R., Lakkala, K. and Sutinen, M.-L.: Effect of the
temperature and the exclusion of UVB radiation on the phenolics and iridoids in *Menyanthes*

*trifoliata L.* leaves in the subarctic, Environmental Pollution, 157, 3471-3478,
10.1016/j.envpol.2009.06.022, 2009.

Mayer, B. and A. Kylling, Technical note: The libRadtran software package for radiative
transfer calculations – description and examples of use, Atmos. Chem. Phys., 5, 1855–1877,
2005

McKinlay, A. F. and Diffey, B. L.: A reference action spectrum for ultraviolet induced
erythema in human skin. CIE Research Note, CIE-Journal, Vol. 6, No.1(17-22), 1987.

Meinander, O., Josefsson, W., Kaurola, J., Koskela, T. and Lakkala, K.: Spike detection and
correction in Brewer spectroradiometer ultraviolet spectra, Opt. Eng. 42, 6, 1812–1819, 2003.

Mitchell, B.G., "Action Spectra for ultraviolet photoinhibition of Antarctic phytoplankton and
a model of spectral diffuse attenuation coefficients", In Response of Marine Phytoplankton to
Natural Variations in UV-B Flux, (Edited by G. Mitchell, I. Sobolev and O. Holm-Hansen),
Proc. of Workshop, Scripps Institution of Oceanography, La Jolla, CA, April 5, 1990

Munakata, N., Kazadzis, S., Bais, A.F., Hieda, K., Rontó, G., Rettberg, P., and Horneck, G.:
Comparisons of spore dosimetry and spectral photometry of solar-UV radiation at four sites in
Japan and Europe. Photochem. Photobiol. 72, 739-745, 2000.

Mäkelä, J.S., Lakkala, K., Meinander, O., Kaurola, J. Koskela, T., Karhu, J.M., Karppinen,
T., Kyrö, E., Leeuw, G. and Heikkilä, A.: In search of traceability: Two decades of calibrated
Brewer UV measurements in Sodankylä and Jokioinen, Geosci. Instrum. Method. Data Syst.
Discuss., doi:10.5194/gi-2015-40, in review, 2016.

Savastiouk, V: A database implementation of data analysis and quality control for the Brewer,
presentation at the 13th Brewer User Workshop. Beijing, China September 12-16, 2011
(https://www.wmo.int/pages/prog/arep/gaw/documents/13th_Brewer_d2_Savastiouk-
Database.pdf)

SCI-TEC Instruments Inc.: Brewer MKII Spectrophotometer, operator's manual, Saskatoon,
Sask. , Canada, 1999.

Setlow, R.B.: The wavelengths in sunlight effective in producing skin cancer: A theoretical
analysis. Proc.Natl.Acad.Sci., U.S.A. 71:3363-3366, 1974.

Slaper, H., Reinen, A. J., Blumthaler, M., Huber, M., and Kuik, F.: Comparing ground-level
spectrally resolved solar UV measurements using various instruments: A technique resolving
effects of wavelength shift and slit width, Geophys. Res. Lett., 22, 2721–2724, 1995.

Webb, A., Gardiner, B., Leszczynski, K., Mohnen, V. A., Johnston, P., Harrison, N. and
Bigelow, D.: Quality Assurance in Monitoring Solar Ultraviolet Radiation: the State of the
Art. World Meteorological Organization (WMO), Global Atmosphere Watch Report, 146,
2003.

Webb, A. R., Gardiner, B. G., Martin, T. J., Leszczynski, K., Metzdoff, J., Mohnen, V. A. and
B. Forgan, Guidelines for site quality control of UV monitoring, Rep.Ser. 126, Environ.
Pollut. Monit. And Res. Programme, World Meteorol. Organ., Geneva, 1998.

WMO (World Meterorological Organization), Report of the WMO-WHO Meeting of Experts
on UVB Measurements, Data Quality and Standardisation of UV indicies, Global Atmopshere
Watch Report No. 95, 1997.

[Figure]

Figure 1. View of Sodankylä observatory. The Brewer is located on the roof in the enclosure on the left.

[Figure]

Figure 2. View of Jokioinen observatory. The Brewer is located on the roof in the enclosure
on the left.

[Figure]

Figure 3. Schematic presentation on the data flow of Brewer UV irradiance data from the

Level 0 raw counts to Level 1 and Level 2 irradiance data.

[1]Meinander at al. 2003; [2]Eq. 1; [3]Eq. 2; [4]Eq. 3; [5]Eq. 4; [6]Eq. 4; [7]Lakkala et al. 2008; [8]Slaper et al. 1995.

[Figure]

Figure 4. Example of Level 0 raw count data recorded by the Brewer #107 on 20 May 2007 at

13:20:07–13:25:18 UTC (on primary vertical axis) and Level 1 / Level 2 responsivities of the instrument for the same day (on secondary vertical axis).

[Figure]

Figure 5a. Time series of Level 2 responsivity of Brewer #037 at 305 nm.

[Figure]

Figure 5b. Partly overlapping time series of Level 1 and Level 2 responsivities of Brewer

#107 at 305 nm.

[Figure]

Figure 6. Schematic presentation of Brewer NRT data processing.

[Figure]

Figure 7. Examples of the Level 1 and Level 2 Brewer irradiance spectra.

[Figure]

Figure 8a. Number of monthly UV irradiance spectra submitted to the EUVDB measured by the

Brewer #037 during 1990-2014.

[Figure]

Figure 8b. Number of monthly UV irradiance spectra submitted to the EUVDB measured by the

Brewer #107 during 1995-2014.

[Figure]

Figure 9. Action spectra used routinely in the processing to derive weighted dose rates from Brewer

UV measurements.

[Figure]

Figure 10. Example of the UVA extension to a spectrum and calculation of the corresponding dose rate weighted with the CIE erythemal action spectrum.

Table 1. Information on the case UV scan of Jokioinen Brewer #107 selected for graphical
demonstration on the phases in the data flow.

| | |
|---|---|
| Date | 20 May 2007 |
| Time of scan | 13:20:07 – 13:25:18 UTC |
| SZA during the scan | 51.3° – 51.4° |
| Total ozone column | 325 DU |
| Visibility | 30 km |

Table 2. Action spectra used routinely in the processing to derive weighted dose rates from Brewer UV

measurements.

| Action spectrum | Reference |
| --- | --- |
| Erythema (CIE weighting function) | McKinlay et al. 1987 |
| Skin cancer in mice corrected for human skin, 299 nm normalization () | de Gruijl, F.R. and J.C. van der Leun, 1994 |
| UVB non weighted 290-320 nm | - |
| UVB non weighted 290-315 nm | - |
| UVA non weighted 315-400 nm | - |
| Generalized plant response (normalised at 300 nm) (Caldwell et al. 1986) | Caldwell et al. 1986 |
| DNA damage (Setlow 1974) | Setlow 1974 |
| Photosynthesis inhibition  (Mitchell 1990) | Mitchell 1990 |
| Previtamin D | CIE 2006 |

---

## Author Comment (AC2) · 27 Apr 2016

Authors' response to the review of the manuscript "Data flow of spectral UV measurements at Sodankylä and Jokioinen" by Mäkelä et al.

The Authors appreciate the constructive comments of the Referee #2 and respond here below to each remark. The comments/questions presented by the referee are indicated as C. The answers are indicated as A. The manuscript has been upgraded following the referee's comments. The corresponding changes in the manuscript are indicated as U.

The revised manuscript has been uploaded as supplement.

C: This paper does not provide any scientifically sound results. It contains a pure outline of the procedures followed for the UV measurements at the two FMI stations but it does not contain any details of the methods or algorithms used in these procedures, or any data. In its present form, it can serve only as a source of general information, like a web site. It is true that detailed descriptions of sites and procedures and different steps in the data processing chain have been published elsewhere, but just a listing of procedures and referencing other papers cannot justify the publication of the manuscript. I think it must be further expanded and structured in a way that could be useful and applicable to other, for example newly established, stations. Furthermore, I think that it could be useful to show at least some sort of time series with measurements at the two sites.

A: The manuscript has been considerably expanded and restructured with an objective to provide useful information in an applicable form to other stations. Description on the steps taken in the processing chain and the phases the data go through has been added, as well as details on the methods and algorithms employed. Three distinct levels of data have been identified and defined. In addition, the phases of the data flow and the outcome of the phases are illustrated using a case scan from Jokioinen.

U: The expanded manuscript is now restructured as follows: 3. Dataflow 3.1. UV data acquisition 3.2 IDEAS - A quality control tool 3.3 UV data processing 3.3.1. Calibration with Level 1 and Level 2 responsivities 3.3.2. Processing algorithms 3.3.3 Online processing - Level 1 data 3.3.4 Offline processing - Level 2 data 3.3.5 Products A case scan from Jokioinen is described in Table 1. This scan is used to illustrate the phases the data go through from the raw counts (Level 0) to calibrated (Level 1 and Level 2) irradiances and further derived dose rates. Time series of the responsivities of the instruments at a selected wavelength (305 nm) are shown.

Specific comments

C: 59-68: Are these statements supported only by "Groebner personal communication"? There have not been published elsewhere?

A: The text has been changed to: The difficulty of the absolute calibration was already seen in intercomparison campaigns of the 1990's (Josefsson et al. 1994, Koskela et al. 1997) and in twenty-first century (e.g., Bais et al. 2001), in which the range of the deviations from the reference for UV spectra was up to ±20%. Despite the efforts to homogenize measurements, in the last European Brewer comparison organized by the COST 1207 project in El Arenosillo, Spain, six Brewers out of 18 differed by more than 10% from the reference, when using the calibration provided by the operator (http://www.pmodwrc.ch/wcc_uv/wcc_uv.php?topic=qasume_audit). The differences are most likely due to slightly different data correction and data processing procedures (for example different procedures to correct for temperature and angular response). Since so many corrections have to be made, small variations can lead to large differences in the outcome."

C: 131: Is there any reference where the cosine correction procedure is described?

A: There is a reference to the procedure available.

U: The reference has been added. "This information and the total ozone calculated by the Brewer is used in the cosine correction procedure (Lakkala et al. 2008)."

C: 131: AWS sampling is ten minutes but the flowchart in Figure 4 sates 5 min. Which of the two is correct?

A: AWS sampling is ten minutes, and the data is uploaded to the FMI climate database. The data is downloaded from the database whenever a new UV scan is transferred to the central server.

U: The Figure 6 has been updated and the text modified accordingly.

C: 136: Similarly, any reference that describes the use of SL-501 for Brewer QA? Otherwise some more information should be added.

A: There is a reference available.

U: The reference has been added and some text added: "These measurements are used in offline quality assurance (QA) procedures to identify erroneous measurements, and to obtain information on changes in the cloud cover (Lakkala et al. 2008)."

C: 151 The paper is about UV measurements. The discussion here and Figure 5 are for total ozone. I suggest to remove both.

A: The authors agree.

U: The discussion has been removed.

C: 151-157: The stability of the UV measurements should be mainly assessed by the 1kW calibration lamps which are not mentioned here, and the 50 W lamps are mainly supporting the assessment of shorter term variations.

A: The authors agree.

U: A new section 3.3.1. "Calibration with Level 1 and Level 2 responsivities" has been added to the manuscript.

C: 183: Most spectral quantities extend beyond the spectral range of the measurements. How is this taken into account and what are the uncertainties involved?

A: We have added some discussion and a clarifying figure in Section 3.3.5.

U: The following text on the method for extending the spectrum has been added in section 3.3.5: "For the calculation of the dose rates requiring integration beyond the upper wavelength limit of the Brewer, the measured spectra are extended using a pre-defined reference UVA spectrum. The extension is adjusted onto the level of the measured spectrum by linear conversion. The ratio of the measured irradiance to the reference irradiance at selected wavelength is used as a scaling factor. For Brewer #037, the wavelength of 324 nm, and for Brewer #107 and #214, the wavelength of 361 nm is used as a point of adjustment." In addition, the following discussion on the uncertainties involved has been included: "All action spectra in routine processing approach

zero towards the longer UVA wavelengths. This means that the uncertainty caused by the artificial UVA extension to the computed dose rate is of the order of 10-3. For the unweighted UV and UVA dose rates, our investigation based on a radiative transfer model simulation suggest uncertainties as high as approx. 2 % caused by the constant scaled UVA extension. This finding is in line with the result obtained by Fioletov et al. 2004."

C: 185-195: The list of different quantities could be supported by a figure with action spectra and a description of how these effective does are calculated. Of course all these are already published elsewhere, but for the completeness of the presentation it would be useful to be included.

A: The authors agree.

U: We have added a description on how the dose rates are calculated in section 3.3.5. A figure on the different action spectra in routine use is also added.

C: 220-221: Again the paper is focused on UV measurements. It could discuss briefly ozone, but showing a figure for ozone (Figure 8) is too much. I suggest removing this Figure.

A: We agree.

U: The figure has been removed.

C: 369: The quality of the flow chart could be improved

A: We agree.

U: The flow chart has been upgraded.

C: 386: The quality of the flow chart is very poor.

A: Indeed, the quality of the flow chart can be substantially improved.

U: The flow chart has been upgraded.

Please also note the supplement to this comment:
http://www.geosci-instrum-method-data-syst-discuss.net/gi-2015-42/gi-2015-42-AC2-supplement.pdf